# Investigation of the Impact of Endodontic Therapy on Survival among Dialysis Patients in Taiwan: A Nationwide Population-Based Cohort Study

**DOI:** 10.3390/ijerph18010326

**Published:** 2021-01-05

**Authors:** Chih-Chien Chiu, Ya-Chieh Chang, Ren-Yeong Huang, Jenq-Shyong Chan, Chi-Hsiang Chung, Wu-Chien Chien, Yung-Hsi Kao, Po-Jen Hsiao

**Affiliations:** 1Division of Infectious Disease, Department of Internal Medicine, Taoyuan Armed Forces General Hospital, Taoyuan City 325, Taiwan; calebchiu.tw@gmail.com; 2Division of Infectious Disease and Tropical Medicine, Department of Internal Medicine, Tri-Service General Hospital, National Defense Medical Center, Taipei 114, Taiwan; 3Division of Nephrology, Department of Internal Medicine, Taoyuan Armed Forces General Hospital, Taoyuan City 325, Taiwan; ajie1124@gmail.com (Y.-C.C.); jschan0908@yahoo.com.tw (J.-S.C.); 4Division of Nephrology, Department of Internal Medicine, Tri-Service General Hospital, National Defense Medical Center, Taipei 114, Taiwan; 5Department of Periodontology, School of Dentistry, Tri-Service General Hospital, National Defense Medical Center, Taipei 114, Taiwan; ndmcandy@mail.ndmctsgh.edu.tw; 6School of Public Health, National Defense Medical Center, Taipei 114, Taiwan; g694810042@gmail.com (C.-H.C.); chienwu@ndmctsgh.edu.tw (W.-C.C.); 7Department of Medical Research, Tri-Service General Hospital, Taipei 114, Taiwan; 8Department of Life Sciences, National Central University, Taoyuan City 320, Taiwan; ykao@cc.ncu.edu.tw

**Keywords:** root canal therapy, dialysis, end-stage renal disease, survival

## Abstract

Objectives Dental problems occur widely in patients with chronic kidney disease (CKD) and may increase comorbidities. Root canal therapy (RCT) is a common procedure for advanced decayed caries with pulp inflammation and root canals. However, end-stage renal disease (ESRD) patients are considered to have a higher risk of potentially life-threatening infections after treatment and might fail to receive satisfactory dental care such as RCT. We investigated whether appropriate intervention for dental problems had a potential impact among dialysis patients. Design Men and women who began maintenance dialysis (hemodialysis or peritoneal dialysis) between January 1, 2000, and December 31, 2015, in Taiwan (total 12,454 patients) were enrolled in this study. Participants were followed up from the first reported dialysis date to the date of death or end of dialysis by December 31, 2015. Setting Data collection was conducted in Taiwan. Results A total of 2633 and 9821 patients were classified into the RCT and non-RCT groups, respectively. From the data of Taiwan’s National Health Insurance, a total of 5,092,734 teeth received RCT from 2000 to 2015. Then, a total of 12,454 patients were followed within the 16 years, and 4030 patients passed away. The results showed that members of the non-RCT group (34.93%) had a higher mortality rate than those of the RCT group (22.79%; *p* = 0.001). The multivariate-adjusted hazard ratio for the risk of death was 0.69 (RCT vs. non-RCT; *p* = 0.001). Conclusions This study suggested that patients who had received RCT had a relatively lower risk of death among dialysis patients. Infectious diseases had a significant role in mortality among dialysis patients with non-RCT. Appropriate interventions for dental problems may increase survival among dialysis patients. Abbreviations: CKD = chronic kidney disease, ESRD = end-stage renal disease, RCT = root canal therapy.

## 1. Introduction

The marked disparities in the incidence and prevalence of treated end-stage renal disease (ESRD) worldwide are well known. Through the United States Renal Data System 2019 Annual Data Report, as seen over the past decade, Taiwan still reported the highest treated ESRD incidence worldwide, with a rate of 504 patients per million general population (PMP). Concurrently, the highest treated ESRD prevalence has also been reported in Taiwan to be 3480 PMP [1]. It is recognized that diabetic nephropathy is the leading cause of ESRD in both developed and developing countries. A case-control study in Taiwan performed by Tsai et al. [2] indicated that a history of hypertension, diabetes, low socioeconomic status, and regular use of folk remedies or over-the-counter Chinese herbs were crucial risk factors for ESRD. Progressive chronic kidney disease (CKD) can cause several complications in other parts of the body with a higher prevalence and intensity as kidney function decreases, including cardiovascular events, anemia, mineral and bone disorders, fluid retention, metabolic acidosis, electrolyte disturbances, and infectious diseases [3]. The leading cause of death in both dialysis and transplanted patients is cardiovascular disease, with the ratio reported to be 10- to 20-fold higher than that observed in populations with normal kidney function. Nevertheless, there are still many patients who die from non-cardiovascular causes, with infectious diseases and malignancies being the most common [1,4,5,6].

However, while clinicians focus on the complications described above, oral health and dental problems are easily neglected. Otherwise, a poor oral condition and its intricate consequences are closely associated with the incidence and progression of CKD. Dental caries and periodontal inflammation affect several different populations, such as pregnant women [7]. Previous studies have also demonstrated excessive rates of oral pathology in CKD patients with one or more oral complications, such as periodontitis, xerostomia, mucositis, and enamel hypoplasia [8,9,10,11]. These symptoms may worsen with declining renal function, resulting in systemic inflammation and malnutrition. CKD has been proven to be a potential mortality risk because of its accelerating impact on cardiovascular disease. Although more adults with CKD suffer from oral disorders than the general public, they may instead receive fewer dental services. A multinational cohort study by Palmer et al. [12] indicated that poor oral health was linked to early death in dialysis patients, whereas preventive dental health practice was predominant in longer survival. Endodontic therapy, also known as root canal therapy (RCT), is indicated when the pulp or soft tissue inside a tooth is damaged by bacterial infection [13]. RCT is a complex therapeutic procedure involving the removal of damaged pulp, cleaning of the infection, and filling the emptied space, and it is classified as a high-cost medical treatment in Taiwan. However, the low socioeconomic status followed by the difficulties in starting a job among hemodialized patients result in the neglect of oral hygiene [14]. Currently available evidence has demonstrated that oral infection may be the site of origin for the dissemination of pathogenic organisms, as their products enter the bloodstream to travel to distant body sites and thus affect the course and pathogenesis of cardiovascular disease, bacterial pneumonia, diabetes mellitus, and low birth weight [15]. Furthermore, hemodialysis and reductions in oral fluid intake can decrease the salivary flow rate and result in changes in oral mucous membranes, leading to xerostomia and increased calculus formation, which add to the risk for microbial infection [16].

We will present an observational retrospective cohort study aimed to investigate the potential effect of receiving appropriate root canal therapy on survival probability in dialysis patients.

## 2. Materials and Methods

### 2.1. Database

Taiwan instituted compulsory universal National Health Insurance (NHI) in 1995, and the health insurance coverage rate currently exceeds 99% of residents. The NHI database is one of the largest and most complete population-based datasets in the world, collecting information from all insured persons, including detailed health services, such as medications, disease diagnoses, examinations, and treatments. The accuracy and validity of the diagnoses have been well demonstrated [17,18,19,20,21]. This study used outpatient and inpatient data, as well as demographic data. Approval for this study was provided by the Institutional Review Board of the Tri-Service General Hospital (TSGH), National Defense Medical Center (Approval No: TSGHIRB-B-109-37).

### 2.2. Definition of Dialysis Patients

This retrospective cohort study enrolled patients who began maintenance dialysis (hemodialysis or peritoneal dialysis) between 1 January 2000 and 31 December 2015. Patients on dialysis were defined by their dialysis codes based on outpatient data. In the case of hemodialysis codes, 58001C, 58014C, 58018C, 58019C, 58020C, 58021C, 58022C, 58023C, 58024C, 58025C, 58026C, 58027C, 58029C, 58030B, and 69006C were all sufficient for inclusion. For those with peritoneal dialysis codes, 58001CA, 58002C, 58009A, 58009B, 58010A, 58010B, 58011A, 58011AB, 58011B, 58011C, 58012A, 58012B, 58017B, 58017C, and 58028C were all acceptable. Patients who were younger than 20 years old were excluded. Patients had to have undergone hemodialysis or peritoneal dialysis for at least 3 months.

### 2.3. Definition of Dialysis Patients with RCT

The procedures of RCT included open chamber, root canal cleaning, and enlargement by either rotary or hand instruments, copious irrigation, and obturation by Gutta-Percha. The procedures and quality followed the therapeutic guidelines and were checked and verified by board-certified endodontist. In this study, we used outpatient data to identify maintenance dialysis patients according to their RCT codes and to classify them into two groups: non-RCT (no codes) and RCT (codes 90001C, 90002C, 90003C, 90004C, or 90015C). RCT was performed after the initiation of dialysis. Subjects with RCT were identified when the first tooth received RCT if multiple teeth were treated after beginning dialysis.

### 2.4. Primary Outcome

The primary outcome was all-cause mortality during the period of 1 January 2000 to 31 December 2015.

### 2.5. Definition of Death

Death was defined as withdrawal of the patient from the NHI program.

### 2.6. Definition of Other Variables

Demographic data included sex, age at the initiation of dialysis, and monthly income. There are two types of dialysis: hemodialysis (HD) and peritoneal dialysis (PD). Conditions diagnosed before the start date of dialysis, namely, diabetes mellitus (ICD-9 codes 250.x), hypertension (ICD-9 code 362.11, 401.x–405.x, 437.2), gout (ICD-9 codes 274.0–274.9), congestive heart failure (ICD-9 code 428.0–428.9, 398.91), coronary artery disease (ICD-9 codes 410.xx–414.xx), cerebrovascular accidents (ICD-9 codes 430–438.xx), peripheral artery disease (ICD-9 code 440.0–440.9, 38.13–38.18, 39.22–39.26, 39.28), chronic lung disease (ICD-9 codes 490–496.x, 500–505.x, 506.4x), chronic liver disease (ICD-9 code 571.x), malignancy (ICD-9 codes 140–208), and retinopathy (ICD-9 code 362.x) were identified as comorbidities.

### 2.7. Statistical Analysis

SAS software (SAS System for Windows, version 9.2; SAS Institute, Cary, NC, USA) was used to perform all statistical analyses in this study. Descriptive statistics were used to analyze the demographic data and the distribution of each variable among the study population. Continuous variables were described as the mean ± statistical difference (SD) and were compared using independent t-test analysis of variance. Categorical variables were described as proportions and compared using the chi-square test. The chi-square test was used to compare each variable in the groups of patients with and without RCT. The cumulative proportion of patients with and without RCT was calculated using the Kaplan–Meier method, and the difference in survival was determined by the log-rank test. Univariate and multivariate analyses were performed with the Cox proportional hazards model. To adjust for potential confounding factors in the relationship between comorbidity, multivariate analysis was used. A *p*-value of < 0.05 was considered statistically significant.

## 3. Results

The records in the NHI databank indicated a total of 14,471 patients on dialysis from 2000 to 2015, 12,454 of whom received maintenance dialysis. A total of 5,092,734 teeth in dialysis patients received RCT between 2000 and 2015. We combined the dialysis data with the RCT data and selected the first tooth per patient if multiple teeth received treatment after the start of dialysis. Among the 12,454 patients with maintenance dialysis, 2633 patients received RCT in the follow-up period, whereas 4030 patients died (Figure 1).

### 3.1. Clinical Characteristics

Table 1 presents the distribution of demographic characteristics, comorbidities, and deaths for the non-RCT and RCT groups. The mean patient age was 57.02 years (±14.75) in members of the non-RCT group and 54.93 years (±13.84) in members of the RCT group (*p* < 0.001). In addition, more women were observed to have undergone RCT (*p* < 0.001) than men. Compared with patients in the RCT group, patients in the non-RCT group were more likely to have diabetes mellitus (32.84% vs. 22.18%; *p* < 0.001), hypertension (51.18% vs. 39.95%; *p* < 0.001), hyperlipidemia (15.71% vs. 11.28%; *p* < 0.001), congestive heart failure (5.69% vs. 4.75%; *p* < 0.001), coronary artery disease (12.20% vs. 8.89%; *p* < 0.001), cerebrovascular accident (9.73% vs. 7.48%; *p* < 0.001), and retinopathy (16.70% vs. 11.70%; *p* < 0.001). Furthermore, members of the non-RCT group displayed higher mortality rates than those of the RCT group (34.93% vs. 22.79%; *p* = 0.001). In contrast, malignancy was less common among the patients in the non-RCT group than in the RCT group (3.48% vs. 5.28%; *p* < 0.001).

Table 2 presents the distribution of demographic characteristics and comorbidities associated with death status. The mean ages at death during the 16-year follow-up period were 65.06 (±12.19) and 52.52 years (±13.88) for patients in the death and non-death groups, respectively (*p* < 0.001). Except for patients with peripheral arterial disease, the dialysis group had higher mortality rates for patients with any of the other comorbidities (*p* < 0.001).

### 3.2. Predictors of Survival Rates

We used the Kaplan–Meier method to analyze survival rates for those who did or did not receive RCT, as determined by the log-rank test. Those in the non-RCT group demonstrated a lower survival rate than those in the RCT group (log-rank *p* < 0.001) (Figure 2). Subsequently, univariate and multivariate Cox proportional models were used to adjust the mortality rates in both groups of subjects (Table 3). A significant association was observed between an increased risk of death with older age, diabetes mellitus, coronary artery disease, cerebrovascular accident, chronic lung disease, chronic liver disease, malignancy, and non-RCT status (all *p* < 0.05). Demographic variables (age, monthly income, dialysis type) and comorbidities, including diabetes mellitus, hypertension, hyperlipidemia, gout, congestive heart failure, coronary artery disease, cerebrovascular accidents, peripheral arterial disease, chronic lung disease, chronic liver disease, malignancy, and retinopathy, were entered into the regression analysis. The multivariate-adjusted hazard ratio for the risk of death was 0.69 (RCT vs. non-RCT; *p* = 0.001) (Table 4). Taken together, the results suggest that dialysis patients who undergo RCT may have a lower risk of death than dialysis patients who do not undergo RCT.

### 3.3. Causes of Death

The causes of death are shown in Table 5. Infectious diseases had a significant role in mortality among dialysis patients with non-RCT (45.54% vs. 34.83%; *p* = 0.003).

## 4. Discussion

Oral health and dental problems are easily neglected in patients undergoing maintenance dialysis. The strength of this study is the first population-based cohort design to investigate the potential effect of appropriate root canal therapy (RCT) on survival in these patients, including hemodialysis and peritoneal dialysis. The results of this study may provide a further prevention strategy for dialysis patients with dental problems. CKD is related to poor outcomes, including an excessive risk of kidney failure, cardiovascular disease, and mortality. These associations may result from the presence of biochemical abnormalities such as increased inflammatory factors, endothelial dysfunction, and enhanced coagulopathy. Accompanying the progressive decline in renal function, patients may have various oral symptoms, such as periodontal disease, decay, missing teeth, oral ulcers, gingival bleeding, halitosis, taste disturbances, uremic odors, periapical lesions, calculus formation, and pale mucosae [22]. The consequences of poor oral health may be more severe in CKD patients because of elderliness, frequent comorbidities such as diabetes, concurrent medications, and immunocompromised states. Kshirsagar et al. [23] reported that individuals with a glomerular filtration rate (GFR) of <60 mL/min/1.73 m^2^ were more likely to have severe periodontal disease than healthy individuals (odds ratio, 2.00; 95% confidence interval, 1.19–3.85).

The inflammatory response has impacted CKD as a risk factor for accelerating CKD progression. In fact, poor oral health is also regarded as a source of inflammation and a contributor to infectious diseases. Initially, patients usually present with gingivitis and periodontitis, which may cause systemic inflammation because of the formation of periodontal pockets colonized with Gram-negative anaerobic bacteria. An inflammatory cell infiltrate is recruited into the lesion and secretes proinflammatory mediators. In the Chronic Renal Insufficiency Cohort (CRIC) study, an inverse relationship between biomarkers of inflammation (interleukin-1β, interleukin-1 receptor antagonist, interleukin-6, tumor necrosis factor-α, C-reactive protein, and fibrinogen) and renal function was advocated [24]. Furthermore, malnutrition and uremic toxin accumulation in dialysis patients may induce immune dysfunction through defects in lymphocyte and monocyte function [8]. Chronic inflammation is thought to contribute to several complications, including arteriosclerosis, atherosclerosis, osteoporosis, frailty, diabetes, malignancy, and so on. Otherwise, the inflammatory and oxidative stress responses may worsen endothelial dysfunction in ESRD patients, accelerate coronary plaque formation, and increase cardiovascular risks [6,25]. Concerning the comorbidities of cardiovascular disease and diabetes mellitus, there was a significant difference (*p* value < 0.05) in our study, which might have resulted from the excessive inflammatory status in patients with poor dental health who did not receive RCT for improvement.

Similar to periodontal inflammation, microbial infection in the root canal system and the periradicular area is prevalent in patients with CKD and ESRD [26]. Persistent root canal infections, such as apical periodontitis, may contribute to infected microorganisms and inflammatory mediators invading the dentinal tubules, accessory canals, and broken-down periapical tissues. Common systemic infectious diseases originating from endodontic infection include bacteremia, endocarditis, aspiration pneumonia, and membranous candidiasis. Chronic bacterial infection is also an important risk factor for atherosclerotic complications and thromboembolic events, which increase the incidence of coronary heart disease [27]. Infection, bacteremia, and sepsis are crucial sources of morbidity and mortality in ESRD patients receiving chronic renal replacement therapy. A previous study disclosed that mortality caused by infection in individuals with both a decreased estimated glomerular filtration rate and an elevated albumin to creatinine ratio was approximately 100- to 300-fold higher for chronic dialysis patients than for the general public [28]. A more advanced proportion of hospital-acquired infections is also noted in dialysis patients due to long-term dialysis-related exposure. When dialysis patients suffer from concomitant dental infection, a complete cycle of antibiotic treatment should be administered. Penicillin, cephalosporins, clindamycin, and their derivatives are safer choices for these patients [29]. In contrast to many studies, in our study, infectious disease was the leading cause of death. We found that more people died from infectious diseases in the non-RCT group (1562 in 3430 patients, 45.54%) than in the RCT group (209 in 600 patients, 34.83%). Furthermore, a tendency for dialysis patients without RCT having a higher all-cause mortality rate (34.93% in the non-RCT group and 22.79% in the RCT group) was also observed.

Lower socioeconomic status (SES) is a risk factor for CKD and progression to end-stage renal disease [30]. Aside from the cross interaction between dental problems and the decline in renal function mentioned above, oral health is usually a lower priority for low SES families, especially those who are struggling with economic hardships and other medical needs. Toothbrushes and toothpaste are the most commonly used oral hygiene products and represent the most effective way to clean teeth. Previous studies have demonstrated that patients with upper SES tend to have sufficient cleaning aids for personal oral hygiene, whereas this is lacking among patients with a lower SES [31,32,33]. The potential benefit of dental health is reflected in survival, cardiovascular events, and quality of life, which might be significant in adults with CKD compared with other settings due to the more severe spectrum of oral disease and infrequent use of dental care practices [17]. In Taiwan, root canal treatment is a high-cost technique and is only partially covered by health insurance. We found that patients with a low SES were more prone to having a higher mortality rate in our study (23.42% in the no income group; 15.21% in the NT$1–15,840 group; 50.25% in the NT$15,841–25,000 group; and 11.12% in the ≥NT$25,001 group), which infers that oral health might be more easily neglected in patients with a low SES than in those with a medium or high SES. Even if they discover these dental problems, they would most likely choose low-cost procedures, such as tooth extraction, instead of high-cost techniques, such as RCT. Tooth loss may increase infectious agents in oral health, accelerate atherosclerosis progression, disturb plaque stability, and cause excessive malnutrition, and it may be considered a potential risk marker for all-cause mortality [34]. A systematic review by Koka and Gupta also reported that a reduced tooth count was associated with higher mortality [35]. However, a recent retrospective case-control study recommended that radical dental interventions to chronic oral infections could be postponed until post-hematopoietic stem cell transplantation [36].

Several previous studies have advocated that poor oral health may be an important risk factor for all-cause mortality in dialysis patients. Although patients in the RCT group were slightly younger (54.93 ± 13.84) than those in the non-RCT group (57.02 ± 14.75), there was still a significant difference in survival probability between these two groups. Through the multivariate regression analysis for the risk of death, the history of diabetes mellitus, coronary artery disease, cerebrovascular accident, chronic lung disease, chronic liver disease, and malignancy all have greater hazard ratios than that of the difference of age. In the other hands, receiving RCT plays a protective role in this analysis. At the same time, for those who without RCT, infectious disease is the leading cause of mortality, whereas these comorbidities mentioned above only have a minor effect. To the best of our knowledge, this is the first retrospective cohort study to explore whether the mortality rate decreases in dialysis patients receiving RCT.

## 5. Limitations

The strengths of this study are its population-based research, use of well-established cohort data with a large sample size, and extended follow-up period to investigate the impact of RCT on survival among dialysis patients. However, this study has several limitations. First, claims data were identified from the NHI database under the principal payment code for the dental service, and complete dental examinations were not performed during face-to-face interviews; however, the decision criteria and ICD-9 coding system for subjects of dental intervention, such as dental filling, periodontal therapy, and RCT, are judged by clinicians. Second, our collected data may still have minor inaccuracies. The accuracy of these data from the NHI database is improved by a cross-checking system with full review by specialists. Thus, these inaccuracies would have a minimal impact on the results. Third, some clinical and laboratory data, such as blood pressure, glycemic control, Kt/V, body mass index, albumin, blood urea nitrogen, creatinine, and inflammatory status, could not be obtained from our database. Meanwhile, the relevance of superior survival probability to receiving RCT and a meaningful impact of infectious disease leading to mortality among the dialysis patients without optimal RCT were revealed in our study. Further prospective studies were needed to identify the clinical or laboratory markers for prognosis or to analyze the infectious episode occurring secondary to dental problems. Regardless of these limitations, our study still has several strengths, including the important advantage of relying on real-world population-based data, a relatively substantial sample size, and the analysis of risk factors and survival results of dialysis patients in Taiwan.

## 6. Conclusions

Our study contributes to filling the knowledge gap, showing relatively better survival in the RCT group than in the non-RCT group among dialysis patients, including hemodialysis and peritoneal dialysis patients. This suggests that we should also never neglect the importance of oral health and endodontic disease in these high-risk patients. Appropriate interventions for dental problems may offer advantages. However, a large and prospective study is warranted to clarify the connection between CKD stage and DENT utilization in CKD subjects.

## Figures and Tables

**Figure 1 ijerph-18-00326-f001:**
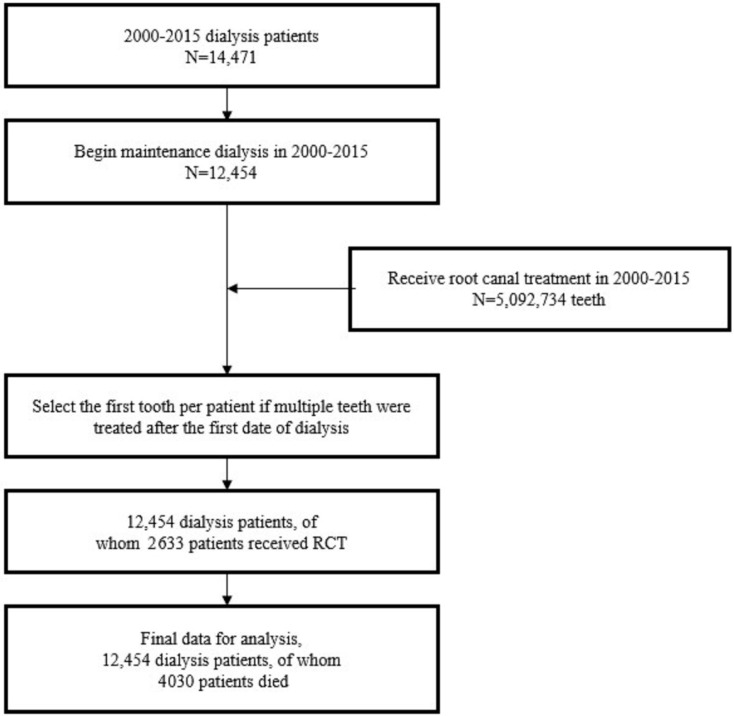
The retrospective cohort study flow diagram.

**Figure 2 ijerph-18-00326-f002:**
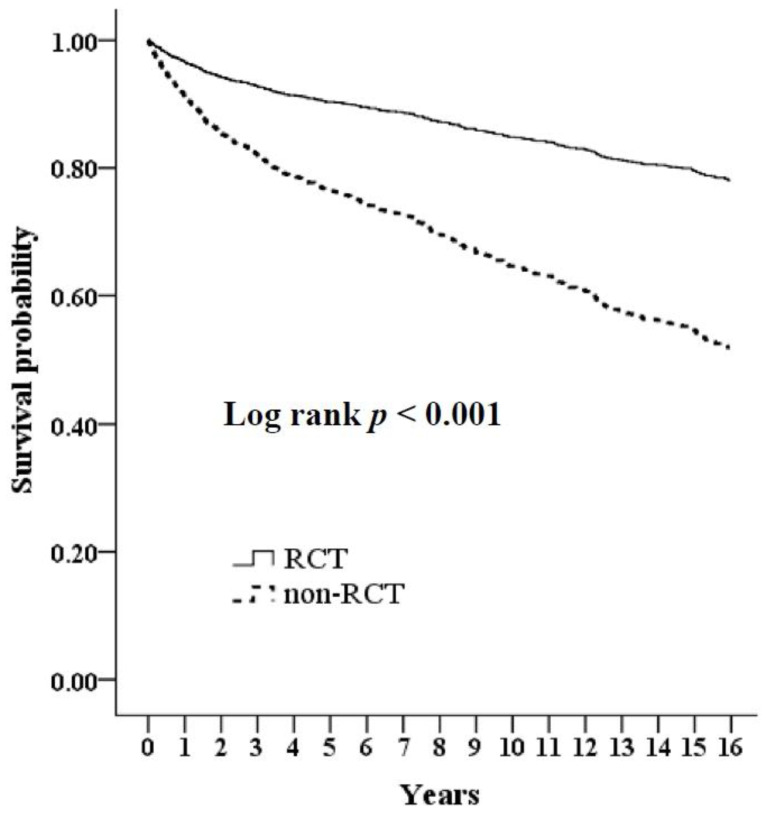
Cumulative survival in dialysis patients with RCT compared to without RCT. The cohort study was followed up for 16 years.

**Table 1 ijerph-18-00326-t001:** Demographic characteristics of dialysis patients with and without root canal therapy (RCT) in the years 2000–2015 (n = 12,454).

Variables	Non-RCT		RCT		*p*
	(n = 9821)		(n = 2633)		
	n	%	n	%	
Age (mean, sd) ^a^	57.02 ± 14.75		54.93 ± 13.84		<0.001 *
Sex					<0.001 *
Female	5009	51.00	1493	56.70	
Male	4812	49.00	1140	43.30	
Monthly income					0.001 *
No income	1942	19.77	473	17.96	
NT$1–15,840	1528	15.56	352	13.37	
NT$15,841–25,000	4917	50.07	1375	52.22	
≥NT$25,001	1434	14.60	433	16.45	
Dialysis type					0.022 *
HD	8920	90.83	2352	89.33	
PD	901	9.17	281	10.67	
Comorbidity					
Diabetes mellitus	3225	32.84	584	22.18	<0.001 *
Hypertension	5026	51.18	1052	39.95	<0.001 *
Hyperlipidemia	1543	15.71	297	11.28	<0.001 *
Gout	1121	11.41	281	10.67	0.285
Congestive heart failure	559	5.69	125	4.75	<0.001 *
Coronary artery disease	1198	12.20	234	8.89	<0.001 *
Cerebrovascular accident	956	9.73	197	7.48	<0.001 *
Peripheral arterial disease	101	1.03	20	0.76	0.263
Chronic lung disease	974	9.92	235	8.93	0.127
Chronic liver disease	782	7.96	192	7.29	0.255
Malignancy	342	3.48	139	5.28	<0.001 *
Retinopathy	1640	16.70	308	11.70	<0.001 *
Death	3430	34.93	600	22.79	0.001 *
Follow-up years (means, sd) ^a^	9.89 ± 2.93		9.71 ± 2.89		0.001 *

* *p* < 0.05; ^a^ Independent-Samples t Test. HD = Hemodialysis; PD = Peritoneal dialysis.

**Table 2 ijerph-18-00326-t002:** Demographic characteristics of dialysis patients who died during 16 years.

Variables	Death		Non-Death		*p*
	(n = 4030)		(n = 8424)		
	n	%	n	%	
Age (means, sd) ^a^	65.06 ± 12.19		52.52 ± 13.88		<0.001 *
Sex					<0.001 *
Female	1990	49.38	4512	53.56	
Male	2,040	50.62	3912	46.44	
Monthly income					<0.001 *
No income	944	23.42	1471	17.46	
NT$1–15,840	613	15.21	1267	15.04	
NT$15,841–25,000	2025	50.25	4267	50.65	
≥NT$25,001	448	11.12	1419	16.84	
Dialysis type					<0.001 *
HD	3703	91.89	7569	89.85	
PD	327	8.11	855	10.15	
Comorbidity					
Diabetes mellitus	2066	51.27	1743	20.69	<0.001 *
Hypertension	2569	63.75	3509	41.65	<0.001 *
Hyperlipidemia	877	21.76	963	11.43	<0.001 *
Gout	601	14.91	801	9.51	<0.001 *
Congestive heart failure	432	10.72	252	2.99	<0.001 *
Coronary artery disease	871	21.61	561	6.66	<0.001 *
Cerebrovascular accident	709	17.59	444	5.27	<0.001*
Peripheral arterial disease	44	1.09	77	0.91	0.379
Chronic lung disease	692	17.17	517	6.14	<0.001 *
Chronic liver disease	478	11.86	496	5.89	<0.001 *
Malignancy	245	6.08	236	2.80	<0.001 *
Retinopathy	1009	25.04	939	11.15	<0.001 *

* *p* < 0.05; ^a^ Independent-Samples t Test. HD = Hemodialysis; PD = Peritoneal dialysis.

**Table 3 ijerph-18-00326-t003:** Hazard ratios of risk of death among dialysis patients with RCT and without RCT.

Variables	Univariate			Multivariate		
	HR	95% CI	*p*	HR	95% CI	*p*
Age	1.09	1.06–1.10	<0.001 *	1.07	1.05–1.09	<0.001 *
Men (Women ^§^)	1.15	0.97–1.34	0.082	1.11	0.91–1.29	0.273
Monthly income (No income ^§^)						
NT$1–15,840	0.80	0.61–1.04	0.062	0.99	0.86–1.31	0.704
NT$15,841–25,000	0.79	0.62–0.93	0.009 *	0.84	0.78–1.24	0.677
≥NT$25,001	0.56	0.40–0.71	<0.001 *	0.79	0.60–1.03	0.125
HD (PD ^§^)	1.23	0.90–1.63	0.172	0.74	0.58–0.97	0.048 *
Diabetes mellitus (yes vs. no)	3.10	2.69–3.63	<0.001 *	1.92	1.53–2.35	<0.001 *
Hypertension (yes vs. no)	2.24	1.83–2.59	<0.001 *	1.12	0.89–1.49	0.569
Hyperlipidemia (yes vs. no)	1.93	1.58–2.35	<0.001 *	1.06	0.72–1.33	0.611
Gout (yes vs. no)	1.56	1.29–1.96	<0.001 *	0.94	0.60–1.35	0.377
Congestive heart failure (yes vs. no)	2.91	2.21–3.72	<0.001 *	1.30	0.99–1.66	0.051
Coronary artery disease (yes vs. no)	3.01	2.53–3.68	<0.001 *	1.59	1.28–1.92	<0.001 *
Cerebrovascular accident (yes vs. no)	2.97	2.31–3.51	<0.001 *	1.42	1.19–1.79	<0.001 *
Peripheral arterial disease (yes vs. no)	1.43	0.65–2.99	0.358	0.90	0.40–1.99	0.835
Chronic lung disease (yes vs. no)	2.64	2.19–3.29	<0.001 *	1.42	1.14–1.75	0.001 *
Chronic liver disease (yes vs. no)	1.95	1.57–2.53	<0.001 *	1.59	1.29–2.03	<0.001 *
Malignancy (yes vs. no)	1.83	1.35–2.49	<0.001 *	1.48	1.18–2.18	0.004 *
Retinopathy (yes vs. no)	2.21	1.82–2.61	<0.001 *	1.20	0.92–1.45	0.077
RCT (vs. non-RCT ^§^)	0.59	0.44–0.79	<0.001 *	0.69	0.51–0.90	0.001 *

* *p* < 0.05; ^§^ ref. HR = hazard ratio; CI = confidence interval; HD = Hemodialysis; PD = Peritoneal dialysis.

**Table 4 ijerph-18-00326-t004:** Hazard ratios of risk of death among dialysis patients with RCT and without RCT.

All-Cause Death	Non-RCT	RCT	95% CI	*p*
Crude HR	1	0.59	0.44–0.79	<0.001 *
Adjusted HR ^a^	1	0.65	0.48–0.83	<0.001 *
Adjusted HR ^b^	1	0.69	0.51–0.90	0.001 *

* *p* < 0.05. HR = hazard ratio; CI = confidence interval; Both crude and adjusted HRs were calculated by Cox proportional hazard regressions. ^a^ Adjustments were made for demographic variables (age, sex, monthly income, and dialysis style); ^b^ Adjustments were made for demographic variables and comorbidities (diabetes mellitus, hypertension, hyperlipidemia, gout, congestive heart failure, coronary artery disease, cerebrovascular accident, peripheral arterial disease, chronic lung disease, chronic liver disease, malignancy, and retinopathy).

**Table 5 ijerph-18-00326-t005:** Cause of death.

Variables	Non-RCT		RCT		*p*
	(n = 3430)		(n = 600)		
	n	%	n	%	
Causes of death					
Coronary artery disease	599	17.46	83	13.83	0.410
Infectious disease	1562	45.54	209	34.83	0.003 *
Cerebrovascular disease	387	11.28	97	16.17	0.189
Malignancy	331	9.65	72	12.00	0.548
Other	551	16.06	139	23.17	0.049 *

* *p* < 0.05.

## Data Availability

Restrictions apply to the availability of these data. Data was obtained from National Health Insurance database and are available from the authors with the permission of National Health Insurance Administration of Taiwan.

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
