# Peer review of "Investigation of the Impact of Endodontic Therapy on Survival among Dialysis Patients in Taiwan: A Nationwide Population-Based Cohort Study"

_ijerph, 2021, doi:10.3390/ijerph18010326_

Round 1

Reviewer 1 Report

Thank you for choosing me as a reviewer of this manuscript.

Abstract section:

Line 31: “decayed caries with infected nerves..” Please change the phrase “infected nerves” to infected pulp or pulp inflammation. I would say that “infected nerves” is colloquial wording. 

Introduction section: I agree with authors that the end-stage chronic kidney disease causes a huge number of complications in many organs and systems. Drug therapies often lead to various pathologies in the oral cavity such as tooth decay, periodontal disease, oral mucosal disease, or decreased saliva secretion. It is very important to mention that inflammatory foci localized in the oral cavity of pregnant women may be a cause of preterm delivery, low birth weight, and preeclampsia, which may endanger both the mother’s and the child’s life. Undoubtedly, hormonal changes that occur during pregnancy cause inflammation, but proper dental plaque control may minimize its risk, consequently limiting labor complications. More articles would be required to confirm these theses- authors may use the articles below:

https://doi.org/10.3390/healthcare8040528

https://doi.org/10.13075/mp.5893.00948

Material and Methods section:

Could authors describe (if they are able) the methods of root canal therapy?. All patients received the same procedure? What what was the main cause of endo treatment (pulp inflammation, necrosis, periapical lesions). Was the endo treatment performed from the National Health System in Taiwan or private? We don’t know anything from the article about the kind of endo treatment. Perhaps authors could describe the procedure more specifically.

Discussion section

In my opinion, the discussion section sufficiently exhausts the problem of the impact of the infection foci localized in the oral cavity on the health of the whole organism. I have only one comment:

Line 287: I would distinguish a separate part from the text called: Limitation of the study

Author Response

Response to the reviewer (Reviewer 1):

  1. Line 31: “decayed caries with infected nerves.” Please change the phrase “infected nerves” to infected pulp or pulp inflammation. I would say that “infected nerves” is colloquial wording.

Remedy: We greatly appreciate this invaluable comment. We changed “infected nerves” to “pulp inflammation” in our revised manuscript. Please see line: 31.

  1. Introduction section: I agree with authors that the end-stage chronic kidney disease causes a huge number of complications in many organs and systems. Drug therapies often lead to various pathologies in the oral cavity such as tooth decay, periodontal disease, oral mucosal disease, or decreased saliva secretion. It is very important to mention that inflammatory foci localized in the oral cavity of pregnant women may be a cause of preterm delivery, low birth weight, and preeclampsia, which may endanger both the mother’s and the child’s life. Undoubtedly, hormonal changes that occur during pregnancy cause inflammation, but proper dental plaque control may minimize its risk, consequently limiting labor complications. More articles would be required to confirm these theses- authors may use the articles below:

https://doi.org/10.3390/healthcare8040528

https://doi.org/10.13075/mp.5893.00948

Remedy: We greatly appreciate your invaluable comment. Dental caries and periodontal inflammation affect several different populations, such as pregnant women [7]. However, the low socio-economic status followed by the difficulties in starting a job among hemodialized patients result in the neglect of oral hygiene [14]. We have added these two references in our revised manuscript. Please see lines: 70-71; lines 83-84.

  1. Katarzynska-Konwa, M.; Obersztyn, I.; Trzcionka, A.; Mocny-Pachonska, K.; Mosler, B.; Tanasiewicz, M. Oral Status in Pregnant Women from Post-Industrial Areas of Upper Silesia in Reference to Occurrence of: Preterm Labors, Low Birth Weight and Type of Labor. Healthcare (Basel). 2020, 8, 528; DOI: 10.3390/healthcare8040528.
  2. Trzcionka, A.; Twardawa, H.; Mocny-Pachońska, K.; Tanasiewicz, M. Oral cavity status of long-term hemodialized patients vs. their socio-economic status. Med Pr. 2020, 71, 279-288. DOI: 10.13075/mp.5893.00948.
  3. Material and Methods section:

Could authors describe (if they are able) the methods of root canal therapy? All patients received the same procedure? What was the main cause of endo treatment (pulp inflammation, necrosis, periapical lesions)? Was the endo treatment performed from the National Health System in Taiwan or private? We don’t know anything from the article about the kind of endo treatment. Perhaps authors could describe the procedure more specifically.

Remedy: We greatly appreciate your constructive and invaluable comments. The causes for these investigated teeth received root canal therapy (RCT) in Taiwan depends on the diagnosis made by board-certified dentists at National Health Insurance (NHI) registered medical institutes, including private practice and hospital-based clinics. The procedure of RCT included open chamber, root canal cleaning and enlargement by either rotary or hand instruments, copious irrigation, and obturation by Gutta-Percha. The procedures and quality should follow the therapeutic guidelines and were checked and verified by board-certified endodontist; otherwise the fee will not be paid. We had emphasized on the procedure more specifically in our revised manuscript. Please see lines: 114-117.

  1. Discussion section

In my opinion, the discussion section sufficiently exhausts the problem of the impact of the infection foci localized in the oral cavity on the health of the whole organism. I have only one comment:

Line 287: I would distinguish a separate part from the text called: Limitation of the study.

Remedy: We greatly appreciate your suggestion. We have made this correction in our revised manuscript. Please see line: 298.

Last, we are deeply honored by the time and effort you spent in reviewing this manuscript. In reviewing and revising our manuscript, we are motivated to read more and thus learn more from your criticisms.

Reviewer 2 Report

This is a report on a retrospective data analysis of dialysis patients. The effect of receiving root canal treatments on patient survival was assessed.

The manuscript is written concisely, and the data generation, presentation and analysis appear to be appropriate. The main question is: are these results credible? I have a really hard time believing that root canal treatment or the lack of the same can impact survival of chronically ill patients. What should be the biological reason for that? Most data collected thus far point in a different direction. Example:

Dissociations of oral foci of infections with infectious complications and survival after haematopoietic stem cell transplantation. Mauramo M, Grolimund P, Egli A, Passweg J, Halter J, Waltimo T.PLoS One. 2019 Dec 18;14(12):e0225099

The authors tried to control confounding by performing a multiple regression analysis. That is fine. However, could it be that it is simply so that overall healthier patients still visit the dentist while unhealthier ones refrain from doing so? Is that somehow reflected in that data set? This should at least be discussed.

Discussion section: start with the main findings, then the limitations, then the comparison to other literature, then outlook and conclusions.

Author Response

 Response to the reviewer (Reviewer 2):

  1. This is a report on a retrospective data analysis of dialysis patients. The effect of receiving root canal treatments on patient survival was assessed.

The manuscript is written concisely, and the data generation, presentation and analysis appear to be appropriate. The main question is: are these results credible? I have a really hard time believing that root canal treatment or the lack of the same can impact survival of chronically ill patients. What should be the biological reason for that? Most data collected thus far point in a different direction. Example:

Dissociations of oral foci of infections with infectious complications and survival after haematopoietic stem cell transplantation. Mauramo M, Grolimund P, Egli A, Passweg J, Halter J, Waltimo T.PLoS One. 2019 Dec 18;14(12):e0225099

Remedy: We are deeply honored by the time and effort you spent in reviewing this manuscript. We have revised the manuscript thoroughly according to your suggestions. The responses to your comments are below. In our study, higher survival probability was noted in the RCT group than that in non-RCT group among the dialysis patients. Infectious disease might be the reason leading to this results. However, further analyses are needed to identify the sort of infection. This is our future work in next step. We have added the above reference in our revised manuscript. Please see lines: 436-438.

  1. The authors tried to control confounding by performing a multiple regression analysis. That is fine. However, could it be that it is simply so that overall healthier patients still visit the dentist while unhealthier ones refrain from doing so? Is that somehow reflected in that data set? This should at least be discussed.

Remedy: We greatly appreciate your constructive and invaluable comments. We explained the difference of mortality between the patient with and without RCT in the last paragraph of discussion: “Through the multivariate regression analysis for the risk of death, the history of diabetes mellitus, coronary artery disease, cerebrovascular accident, chronic lung disease, chronic liver disease, and malignancy all have greater Hazard ratios than the difference of age do. In the other hands, receiving RCT plays a protective role in this analysis. At the same time, for those who without RCT, infectious disease is the leading cause of mortality, whereas these comorbidities mentioned above only have a minor effect.” We have made this correction in our revised manuscript. Please see lines: 290-295.

  1. Discussion section: start with the main findings, then the limitations, then the comparison to other literature, then outlook and conclusions.

    Remedy: Thanks for your suggestions. It is a reasonable structure of an article, but we use the template form of      

    this journal.

Last, we are deeply honored by the time and effort you spent in reviewing this manuscript. In reviewing and revising our manuscript, we are motivated to read more and thus learn more from your criticisms.

Round 2

Reviewer 2 Report

Thank you. You have addressed my concerns adequately.